# Differential microRNA profiling of the Marshallese population in Arkansas reveals a higher association with chronic diseases

Gohar Azhar[1]◉*, Ambika Verma[1]◉, Pankaj Patyal[1], Wei Zhang[2], Shakshi Sharma[1], Patricia E. Savary[1], Sheldon Riklon[3], Philmar Mendoza Kabua[3], Pearl A. McElfish[3], Jeanne Y. Wei[1]

1 Donald W. Reynolds Institute on Aging, Department of Geriatrics, University of Arkansas for Medical Sciences, Little Rock, Arkansas, United States of America, 2 Department of Mathematics and Statistics, University of Arkansas at Little Rock, Little Rock, Arkansas, United States of America, 3 College of Medicine, University of Arkansas for Medical Sciences Northwest, Springdale, Arkansas, United States of America

◉ These authors contributed equally.
* Azhargohar@uams.edu

## Abstract

The Marshallese communities face disproportionately high prevalence of chronic diseases, including diabetes, obesity, cardiovascular disease, and inflammatory conditions. Differences in miRNA expression may contribute to investigate the potential risk of chronic diseases in Marshallese people. In this study, we used RNA isolated from blood samples of Marshallese participants that resides in Arkansas to perform miRNA expression profiling and differential expression analysis. Specifically, blood samples were collected from 47 Marshallese participants after obtaining written informed consent and were subjected to Illumina-based next-generation RNA sequencing. Using the miRBase database, we identified the miRNAs that were most significantly expressed based on log2 fold change values, applying a Bonferroni-corrected $P$-value threshold of $< 0.01$. We found that a total of 63 human miRNAs were differentially expressed in the Marshallese subjects, with 52 miRNAs significantly upregulated and 11 miRNAs downregulated in males compared with females. Notably, 2 miRNA families, hsa-miR-548 and hsa-let-7, were significantly upregulated in the Marshallese population and are known to play important roles in regulating inflammatory responses. Further analysis revealed the 25 miRNAs that had the largest significant difference in expression with a cutoff of 1.5 when comparing males with females. Among these, we observed that 7 miRNAs were upregulated, and 18 miRNAs were downregulated by greater than 1.5 log2 fold change in males versus females. Interestingly, upregulated expression of hsa-miR-548k in males and hsa-miR-496 in females were both associated with diabetes mellitus. Diabetes remains a major health concern in the Marshallese community and may accelerate comorbidities related to cardiovascular conditions and cognitive decline.

which permits unrestricted use, distribution, and reproduction in any medium, provided the original author and source are credited.

**Data availability statement:** All relevant data are within the manuscript and its Supporting information files.

**Funding:** This study was supported by grant R01MD013852 from the National Institutes of Health, and in part, by the Lyon Aging Research Program, Reynolds Institute on Aging, University of Arkansas for Medical Sciences, Little Rock, AR.

**Competing interests:** The authors have declared that no competing interests exist.

Therefore, the specific roles of these miRNAs in relation to these health issues warrant further investigation.

## Introduction

Arkansas is home to one of the largest populations of Marshallese people from the Republic of the Marshall Islands living in the continental US, with an estimated 15,000 Marshallese people residing in the northwest part of the state [1]. The Marshallese community experiences a disproportionately high burden of chronic diseases, including type 2 diabetes mellitus (T2DM), obesity, hypertension, cardio-vascular disease, and various infectious diseases, all of which contribute to elevated morbidity and mortality rates in this population [1–11]. Among these, T2DM poses a particularly urgent public health concern, with prevalence estimates among Marshallese individuals approximately 400% higher than those reported in the general U.S. population [12–14]. In the Republic of the Marshall Islands (RMI), the adult diabetes prevalence reaches 33.8%, significantly exceeding both global (9.3%) and U.S. (13.3%) averages [15]. According to the International Diabetes Federation, the RMI ranks among the countries with the highest age-adjusted diabetes prevalence in adults, estimated at 25.7% in 2024 [16]. Similarly, data from Marshallese adults residing in the United States reveal diabetes prevalence rates of 44.2% in Hawaii and 46.5% in Arkansas, with an additional 25.3% and 21.4% meeting criteria for prediabetes in these respective states [17]. These metabolic disorders frequently coexist with other chronic conditions, further exacerbating health disparities within the Marshallese community [11].

In particular, the coexistence of obesity and diabetes represents a significant concern. Among adults in the RMI, obesity prevalence is alarmingly high, affecting 60.4% of women and 52.2% of men—far surpassing regional averages of 31.7% and 30.4%, respectively [18]. Concurrently, diabetes affects 23.4% of adult women and 22.6% of adult men [18]. These sex-specific disparities in disease prevalence highlight the need to explore the underlying molecular mechanisms contributing to differential susceptibility and progression across a wide range of chronic diseases. Sex-based differences in gene expression are increasingly recognized as critical biological factors that may also influence disease susceptibility, progression, and therapeutic response [19]. Exploring these profiles offers the potential to uncover molecular pathways associated with differential risk or resilience between sexes, thereby support the development of sex-informed diagnostic biomarkers and therapeutic strategies [19].

One way in which gene expression can be regulated is through a subset of non-coding RNAs [20]. Unlike messenger RNAs, which code for proteins, these non-coding RNAs are not translated into proteins; instead, they play crucial roles in regulating various biological processes. One important type of non-coding RNA is microRNAs (miRNA), that are short, endogenous, non-coding RNA molecules (often 18–25 nucleotides), which are highly conserved across species and play crucial roles in disease development [21], likely through regulation of gene expression. miRNAs regulate gene expression by targeting and binding to messenger RNAs of

protein-coding genes, leading to mRNA degradation or inhibition of translation. miRNAs are involved in nearly all biological processes, including cellular development, proliferation, differentiation, maturation, senescence, apoptosis, tumorigenesis, stress response, cell signaling, and cellular interactions [22]. Importantly, altered expression or dysregulation of miRNAs has been associated with a wide range of diseases, highlighting the importance of miRNAs as potential biomarkers and therapeutic targets of disease [21–23]. Thus, understanding miRNAs is crucial for elucidating their roles in both health and disease, providing valuable insights for therapeutic interventions and diagnostics across various pathological conditions like diabetes. By investigating the specific processes regulated by miRNAs and identifying new miRNA targets associated with these diseases, we can pave the way for innovative treatment strategies.

In this study, high-throughput sequencing and bioinformatics analyses were used to elucidate the miRNA expression patterns in peripheral blood collected from Marshallese people in Arkansas. To the best of our knowledge, we are the first to investigate miRNA expression profiles specifically within the Marshallese community. Investigating miRNA profiles in this population can provide valuable insights into disease mechanisms and contribute to more effective health interventions. miRNA studies can provide insights into disease mechanisms and identify potential therapeutic targets. This research aims to uncover unique miRNA signatures that could provide insights into the molecular mechanisms underlying health disparities and chronic diseases prevalent in the Marshallese population.

## Results

### Health survey

A comprehensive health survey was done to all 50 study participants to assess their health status, with full participation achieved (Table 1). The most commonly reported condition was Type 2 diabetes, affecting 53% of participants. Other prevalent conditions included anxiety (31%), depression (29%) and hypertension (26%). Additionally, 10% of participants reported a diagnosis of heart disease, 6% reported Type 1 diabetes, and 2% reported a history of head trauma. These findings highlight a substantial burden of chronic health conditions within the Marshallese study population.

### Sequencing analysis

The sequencing run produced an average of 20.91 million reads per sample (S1A Fig). The median Phred score for all samples was above 30 across the entire read length, indicating high-quality sequencing data (S1B Fig). A PHRED score

Table 1. Demographics of the study subjects.

|  | Demographics | Total Count | (%) |
|---|---|---|---|
| 1. | Sex (n = 50) |  |  |
|  | Female | 36 | 72% |
|  | Male | 14 | 14% |
| 2. | Age |  |  |
|  | 30s & 40s | 34 | 68% |
|  | 50+ | 16 | 32% |
| 3. | Self -Reported Health status |  |  |
|  | Type 2 diabetes (n = 43) | 23 | 53% |
|  | Anxiety (n = 48) | 15 | 31% |
|  | Depression (n = 48) | 14 | 29% |
|  | Hypertension (n = 46) | 12 | 26% |
|  | Heart disease (n = 48) | 5 | 10% |
|  | Type 1 diabetes (n-47) | 3 | 6% |
|  | Head trauma (n = 46) | 1 | 2% |

>30 is considered standard for high-quality sequencing, indicating a base call accuracy of 99.9%. All samples exceeded this threshold across the entire read length, and no samples failed QC at any stage of the pipeline. Reads were processed by trimming the common sequence and keeping reads with lengths of 15–55 nt. Trimmed read length distributions showed a distinct peak at approximately 22 nt, aligning with the expected length of miRNA (S1C Fig). Approximately 1.49 million reads (median across samples) were filtered out prior to deduplication, primarily due to reads being shorter than 15 nt. To remove data deduplication, reads were deduplicated based on their unique molecular identifiers (UMIs). The number of reads per sample were shown together with the number of UMI groups after deduplication (S1D Fig).

## Mapping statistics

Deduplicated reads were mapped to miRBase (release 22, prioritizing Homo sapiens) and piRNA (piRNAdb.hsa.v1.7.6) (Fig 1A). The proportion of reads successfully mapped to the miRNA database ranged from 92% to 95% (Fig 1A). Reads that mapped to miRBase were split by type of match: (i) perfect complementarity to the reference sequence, (ii) isomiRs mapping with a maximum of 2 mismatches to the reference sequence, and (iii) isomiRs with alternative start or end positions (maximum of 2 nt) (Fig 1B). To better characterize small RNA sequences that did not align with known small RNA databases (e.g., miRBase), we performed a secondary alignment step in which the remaining unmapped reads were mapped to the whole human genome via ENSEMBL GRCh38 version 98 annotation (hg38) (S2A Fig). Normalized spike-in signals were assessed for each sample with Spearman correlation. Reads submitted to deduplication were filtered if they did not contain the common sequence or have a UMI with a length other than 12 nt or have lengths of 15–55 nt (S2A-C Fig).The S2A Fig displays the number of reads successfully mapped to the genome after excluding those annotated as known small RNAs. These reads were further categorized based on genomic annotation into different gene elements, including exons, introns, and intergenic regions (S2B Fig). We observed that a significant portion of mapped reads aligned to intronic regions, and a smaller fraction mapped to exons and intergenic regions (S2B Fig). Following alignment

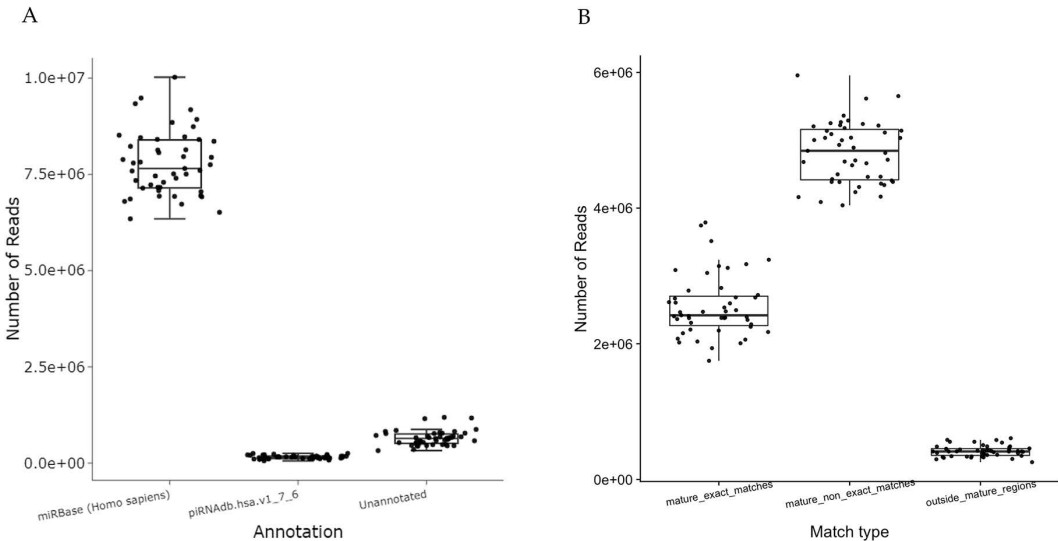

**Fig 1. Mapping statistics of deduplicated reads (*n* = 47 samples). (A)** Number of reads mapped per sample to miRBase, piRNAdb.hsa.v1.7.6, and unannotated categories. The X-axis indicates the annotation source, and the Y-axis shows the corresponding read count intensity across all samples. **(B)** Distribution of reads mapped to miRBase, categorized as: (i) exact matches to mature miRNAs, (ii) non-exact matches with up to 2 mismatches, and (iii) reads mapping outside mature regions. All values are derived from variance-stabilized, deduplicated reads, representing high-confidence alignments.

of unmapped reads to the reference genome, we classified the mapped reads by gene biotype to gain further insights into their origin and potential function based on genome annotation. Across all samples, the proportion of reads mapping to each gene biotype varied but showed consistent patterns with a preference for miscellaneous RNA and long non-coding RNA (S2C Fig). Variation between samples in the relative abundance of specific biotypes may reflect differences in sample origin, RNA integrity, or biological conditions.

## Unsupervised analysis

We used variance-stabilizing transformation (VST) as implemented in DESeq2 to prepare the data for downstream exploratory analyses, such as clustering and principal component analysis (PCA). VST was chosen over alternatives like TPM normalization because it effectively stabilizes the variance across the full range of expression values, particularly addressing the high variability seen in low-abundance transcripts. This makes it more appropriate for the unsupervised analyses performed in this study by which we aimed to identify factors that account for the most variance across the sequenced samples. PCA was used to reduce the dimension of large data sets and to explore sample clusters arising based on the expression profile. The data points that represent the samples are projected onto the 2D plane such that they spread out in the 2 directions that explain most of the differences among males versus females (Fig 2A). The variance-stabilized transformation was performed on the raw count matrix, and 35 genes with the highest variance across samples were selected for hierarchical clustering (Fig 2B). The median absolute deviation was also calculated, and the top 100 genes were selected for hierarchical clustering (S3 Fig).

## Empirical analysis of differential gene expression

For each differential gene expression analysis, the fold change of each gene was plotted against its mean expression among all samples, and the result of the statistical test is represented in an MA plot (Fig 3A). Significant changes are

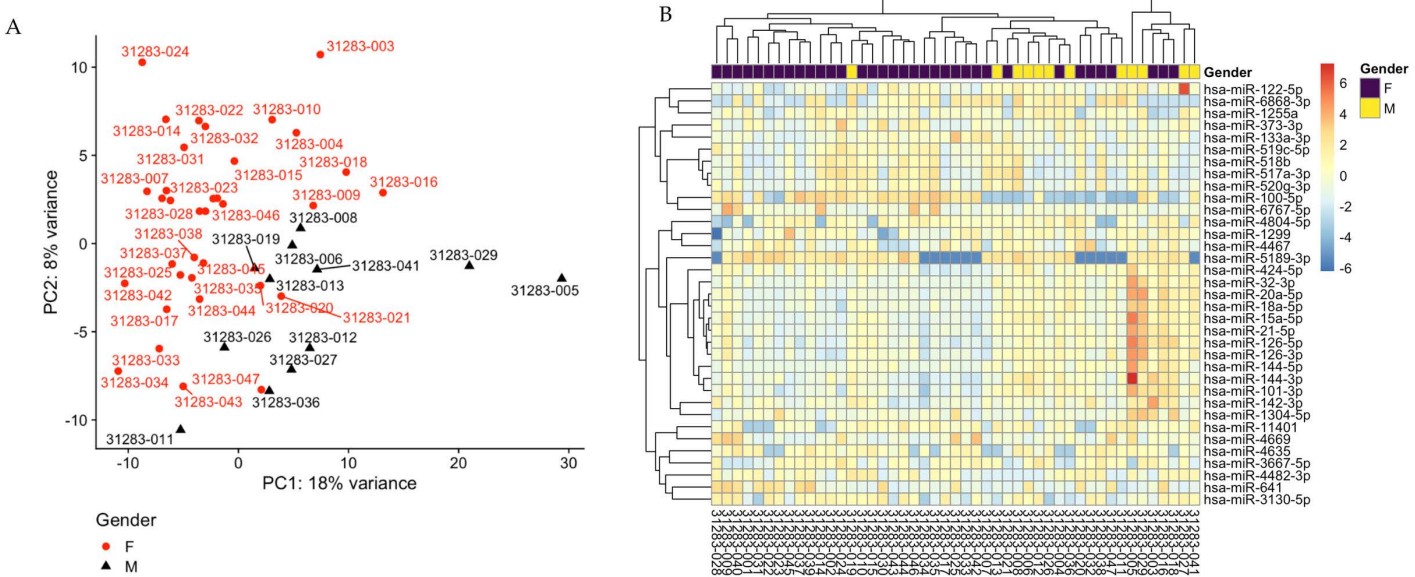

**Fig 2. A variance stabilizing transformation was performed on the raw count matrix. (A)** Principal component analysis of data after a variance stabilizing transformation. **(B)** Hierarchical clustering of the top 35 miRNAs ranked by variance. Each row represents 1 gene, and each column represents 1 sample. The colors represent deviations from the row mean based on variance-stabilized, normalized expression values. Red indicates higher expression relative to the row mean, and blue indicates lower expression.

defined as FDR < 0.01. The adjusted *P*-value cutoff for calling a change significant was kept at *P*adj < 0.01. The differentially expressed miRNAs were screened as per log2 fold change value of expression. A total of 63 human miRNAs were selected, and among them, 52 human miRNAs were significantly upregulated, and 11 miRNAs were significantly downregulated in male versus female subjects (Fig 3A). Further, a variance-stabilized transformation was performed on the raw count matrix, and after statistical testing, the differentially expressed 63 miRNAs were selected for hierarchical clustering (Fig 3B). Through hierarchical clustering analysis, the 2 most abundant human miRNA families: hsa-miR-548 and hsa-let-7, were identified that play important roles in regulating several inflammatory responses (S1 Table).

## Differential gene expression with a 1.5 cutoff point

Fold changes in gene expression of male versus female participants were calculated and transformed on a log2 scale for normalization. A volcano plot was generated with a 1.5 cutoff point for absolute of log2-fold change in expression. Twenty-five miRNAs that had statistically significant differential expression with a cutoff point of 1.5 when comparing males with females were screened. Expression values are shown as normalized counts in counts per million (CPM) and mean represents the average normalized expression across samples within each group. Our findings revealed that 7 miRNAs were upregulated by greater than 1.5-log2 fold change in males versus females, including hsa-miR-885-3p, hsa-miR-548au-5p, hsa-miR-190a-5p, hsa-miR-548k, hsa-miR-18a-5p, hsa-miR-95-3p, and hsa-miR-18b-5p. A total of 18 miRNAs were downregulated by greater than 1.5-log2 fold change in males versus females, including hsa-miR-100-5p, hsa-miR-133a-3p, hsa-miR-6894-3p, hsa-miR-219b-5p, hsa-miR-641, hsa-miR-6767-5p, hsa-miR-6859-3p, hsa-miR-6739-3p, hsa-miR-758-5p, hsa-miR-6736-3p, hsa-miR-508-5p, hsa-miR-188-3p, hsa-miR-1228-5p, hsa-miR-496, hsa-miR-1912-3p,

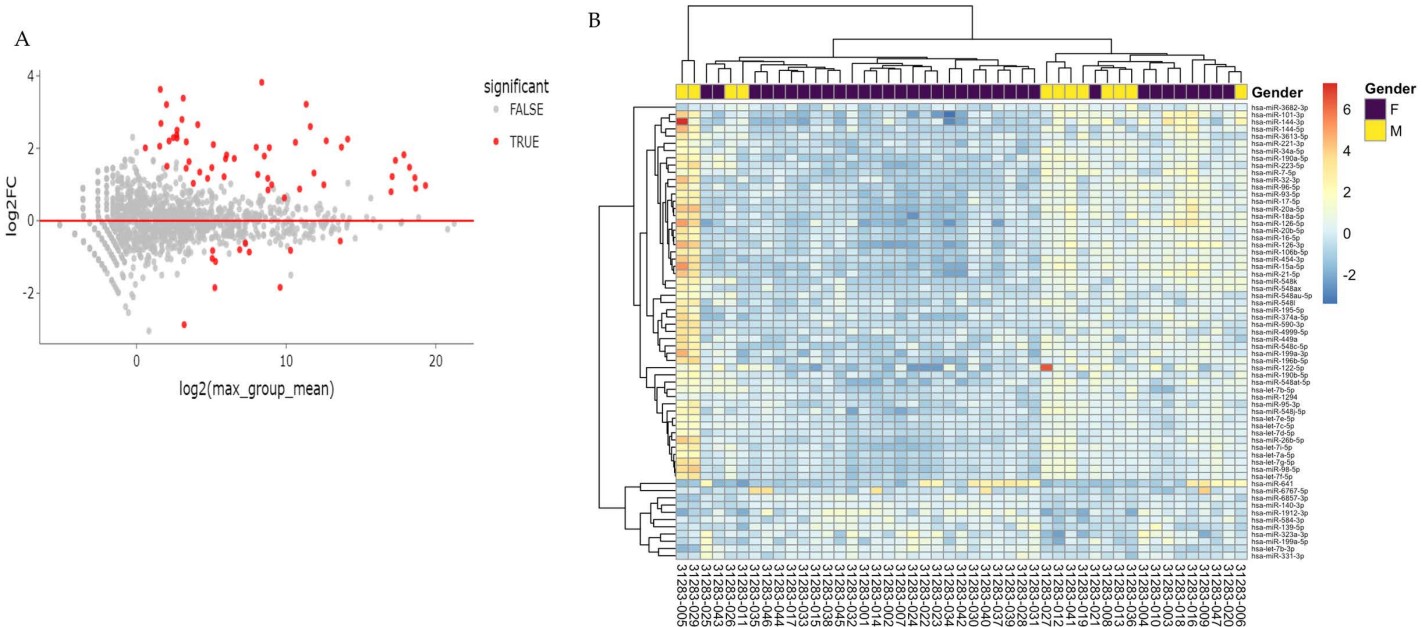

**Fig 3. Differential expression analysis of significantly regulated miRNAs. (A)** Volcano plot displaying differentially expressed miRNAs. Significance was determined using FDR < 0.01 and Bonferroni-corrected p-values. The X-axis represents log2 of the maximum group means, and the Y-axis shows the log2 fold change. Red dots indicate significantly upregulated (above the red line) or downregulated (below the red line) miRNAs. Gray dots indicate miRNAs with no significant differential expression. **(B)** Hierarchical clustering heatmap of 63 significantly differentially expressed miRNAs. Each row represents one miRNA, and each column represents one sample. Colors indicate expression relative to the row mean, based on variance-stabilized, normalized values: red denotes higher expression, and blue denotes lower expression.

hsa-miR-6730-3p, hsa-miR-3125, and hsa-miR-6767-3p (Fig 4A, Table 2). A clustered analysis of the 25 miRNAs that differed significantly between groups revealed the miRNAs profiling in males versus females (Fig 4B).

## Network-based visual analysis

The interactions of 25 differentially expressed miRNAs with a cutoff point of 1.5 were analyzed with miRNet 2.0, and based on the results, differentially expressed miRNAs showed associations with their target diseases (Fig 5). Among the 7 upregulated miRNAs in males versus females, only hsa-miR-190a-5p, hsa-miR-548k, and hsa-miR-885-3p showed associations with several chronic diseases (Fig 5). Among the 18 downregulated miRNAs in males versus females, only hsa-miR-496, hsa-miR-64, hsa-miR-100-5p, hsa-miR-188-3p, hsa-miR-6767-3p, hsa-miR-508-5p, hsa-miR-1228-5p, hsa-miR-3125, and hsa-miR-133a-3p showed associations with chronic diseases (Fig 5). Among the 25 differentially expressed miRNAs, the upregulated expression of hsa-miR-548k in males and hsa-miR-496 in females was associated with diabetes mellitus.

## miRNA target prediction and pathway association

To explore potential regulatory mechanisms, we analyzed predicted target genes of two differentially expressed miR-NAs—hsa-miR-548k, significantly upregulated in males, and hsa-miR-496, upregulated in females—using TargetScan 8.0. The complete information related to hsa-miR-548k, and hsa-miR-496 target predictions and their pathway association are provided as supporting files 1 and 2. For hsa-miR-548k, main targets involved in diabetes-related pathways were identified, including TGFBR1, STAT1, NFKB1, MAPK1, and STAT5A (Table 3). These genes are key components of the TGF-β, JAK/STAT, NF-κB, and MAPK pathways. TGFBR1 exhibited the highest number of predicted binding sites (n = 4), including a high affinity 8mer site. Cumulative context++ scores ranged from –0.26 (TGFBR1) to –0.03 (STAT5A), indicating varying degrees of predicted repression. For hsa-miR-496, two conserved targets—STAT5B (JAK/STAT pathway) and TGFBR2 (TGF-β signaling)—were identified, each with 8mer binding sites and cumulative context++ scores of –0.28 and –0.14, respectively (Table 3). Predicted occupancy values further support the potential regulatory roles of these interactions. Together, these findings suggest that hsa-miR-548k and hsa-miR-496 may modulate key signaling pathways involved in glucose metabolism and diabetes pathogenesis through post-transcriptional repression of critical gene targets.

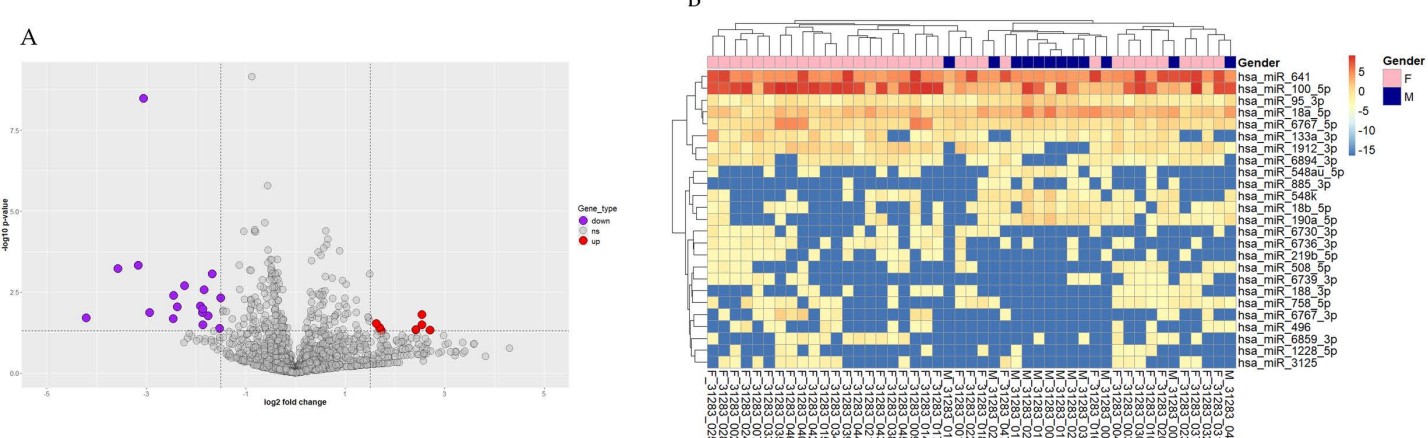

**Fig 4. Fold changes in miRNA expression between males and females. (A)** A volcano plot generated with a cutoff point of 1.5 for the absolute log2 fold change in expression. Red = upregulated miRNAs. Purple = downregulated miRNAs. **(B)** Heatmap of the clustered analysis of identified miRNAs in both groups. Each row represents 1 gene, and each column represents 1 sample. The colors represent deviations from the row mean based on variance-stabilized, normalized expression values. Red indicates higher expression relative to the row mean, and blue indicates lower expression.

**Table 2. List of the top 25 differentially expressed miRNAs in male versus female subjects.**

| | Name | Mean CPM (Females) | Mean CPM (Males) | Log2FC | *P*-value |
|---|---|---|---|---|---|
| **Upregulated miRNAs (Males versus Females)** | | | | | |
| 1 | hsa-miR-885-3p | 0.013920681 | 0.090829 | 2.705917 | 0.047102 |
| 2 | hsa-miR-548au-5p | 0.041636081 | 0.242902 | 2.544468 | 0.032154 |
| 3 | hsa-miR-190a-5p | 0.170198211 | 0.988485 | 2.538003 | 0.01563 |
| 4 | hsa-miR-548k | 0.137960258 | 0.739571 | 2.422436 | 0.045328 |
| 5 | hsa-miR-18a-5p | 4.400958108 | 14.56911 | 1.727023 | 0.048032 |
| 6 | hsa-miR-95-3p | 0.545367562 | 1.760753 | 1.690892 | 0.039983 |
| 7 | hsa-miR-18b-5p | 0.171712779 | 0.530667 | 1.627809 | 0.029565 |
| **Downregulated miRNAs (Males versus Females)** | | | | | |
| 8 | hsa-miR-100-5p | 169.3417687 | 59.84917 | −1.50053 | 0.004704 |
| 9 | hsa-miR-133a-3p | 0.911280614 | 0.315518 | −1.53017 | 0.042095 |
| 10 | hsa-miR-6894-3p | 0.436161352 | 0.136328 | −1.67778 | 0.00087 |
| 11 | hsa-miR-219b-5p | 0.079238537 | 0.023377 | −1.76111 | 0.01703 |
| 12 | hsa-miR-641 | 99.00521214 | 27.66111 | −1.83965 | 0.002658 |
| 13 | hsa-miR-6767-5p | 4.860129097 | 1.338159 | −1.86075 | 0.03237 |
| 14 | hsa-miR-6859-3p | 0.118570131 | 0.032639 | −1.86107 | 0.010333 |
| 15 | hsa-miR-6739-3p | 0.07693119 | 0.020987 | −1.87405 | 0.013447 |
| 16 | hsa-miR-758-5p | 0.181410973 | 0.048204 | −1.91203 | 0.008392 |
| 17 | hsa-miR-6736-3p | 0.131335618 | 0.028015 | −2.22899 | 0.001983 |
| 18 | hsa-miR-508-5p | 0.102252389 | 0.01972 | −2.37437 | 0.008877 |
| 19 | hsa-miR-188-3p | 0.070241253 | 0.012863 | −2.44905 | 0.004015 |
| 20 | hsa-miR-1228-5p | 0.051098261 | 0.00927 | −2.46269 | 0.020749 |
| 21 | hsa-miR-496 | 0.075278396 | 0.00986 | −2.93254 | 0.013372 |
| 22 | hsa-miR-1912-3p | 1.22545952 | 0.147171 | −3.05776 | 3.29E-09 |
| 23 | hsa-miR-6730-3p | 0.113211195 | 0.012617 | −3.16553 | 0.000464 |
| 24 | hsa-miR-3125 | 0.110072643 | 0.00927 | −3.5698 | 0.000592 |
| 25 | hsa-miR-6767-3p | 0.238160965 | 0.012863 | −4.2106 | 0.019636 |

## Discussion

The Marshallese community in Arkansas faces unique health challenges, with the leading causes of death include diabetes-related complications, hypertension, and cancer [24,25],which may be linked to a combination of genetic, environmental, and lifestyle factors [1]. Research into the role of miRNAs in disease development, diagnosis, and prognosis is gaining momentum across various fields. Our research highlights the critical role of miRNAs in the development and progression of various diseases prevalent in the Marshallese community. These miRNAs are essential for maintaining health, and their dysregulation has been associated with a range of diseases, including diabetes, cardiovascular disorders, neurodegenerative diseases, and cancers [26]. Indeed, dysregulation of miRNAs is frequently observed in several human diseases, including vascular complications associated with obesity and diabetes [27].

In this study, we employed next-generation sequencing to profile miRNA expression from peripheral blood samples of Marshallese individuals, uncovering a complex regulatory landscape involving over 2,600 unique miRNAs. Principal component analysis revealed distinct expression patterns between males and females, highlighting sex-specific regulatory differences. Among the 63 differentially expressed miRNAs, families such as miR-548 and let-7 were predominant—both of which are well-documented regulators of gene networks implicated in chronic diseases [28–37]. The observed upregulation of miRNAs like hsa-miR-21, hsa-miR-26, hsa-miR-93, hsa-miR-96, hsa-miR-126, and hsa-miR-496, all of which

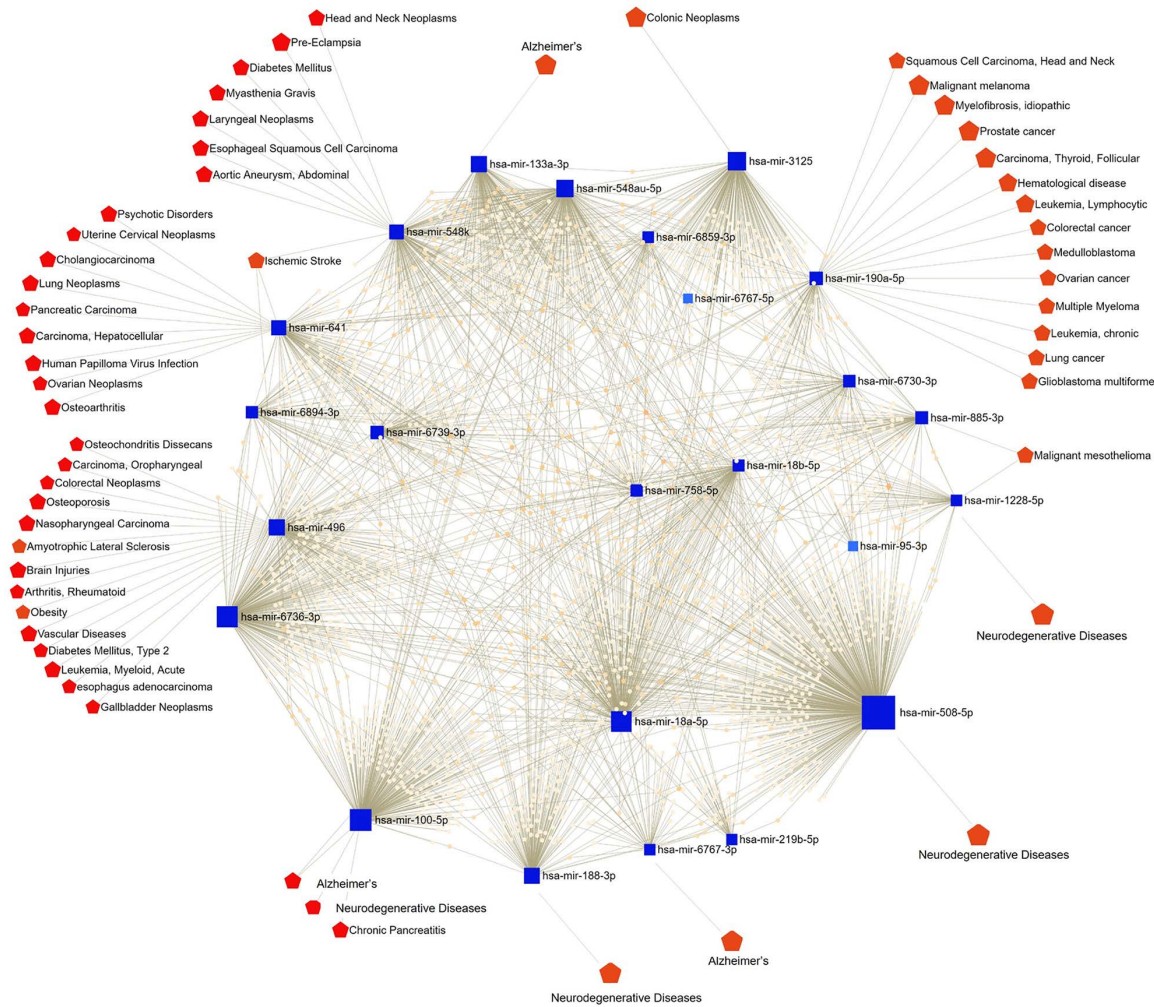

**Fig 5. Functional map showing the miRNA disease-network analysis via miRNet 2.0.** This map represents the interactions between miRNAs (blue) with potentially linked diseases (red). The blue squares represent the miRNA, and red nodes represent the target diseases or disease categories.

have validated roles in insulin resistance, suggests a potential contribution to the molecular pathogenesis of type 2 diabetes [38–43]. Additionally, hsa-miR-496, found differentially expressed in our data, has been shown to modulate components of the PI3K/Akt and TGF-β pathways, both of which are central to glucose homeostasis and fibrotic progression in diabetic complications [43]. Similarly, other miRNAs which were dysregulated, including hsa-miR-17 and hsa-miR-21 have been reported to be associated with hypertension [44,45], hsa-miR-144-3p and hsa-miR-144-5p, have been implicated in neurodegenerative processes such as oxidative stress and tau phosphorylation in Alzheimer's disease [46,47]. Others including hsa-miR-21, hsa-miR-496, hsa-miR-548j-5p, hsa-miR-584-3p, and hsa-let-7a-5p have established associations with cardiovascular diseases [48–53]. These findings reinforce the biological relevance of circulating miRNAs as non-invasive biomarkers and potential mediators to chronic disease. Since miRNAs can regulate nearly all biological processes, their dysregulation is associated with a range of complex diseases and pathological conditions.

High-throughput sequencing technologies have enabled detailed profiling of miRNA expression and their associations with chronic diseases. In our study, 25 significantly differentially expressed miRNAs were identified between males and

**Table 3. Predicted hsa-miR-548k and hsa-miR-496 targets in key diabetes-related pathways.**

| Gene Symbol | Gene Name | Pathway | Total Sites | 8mer | 7mer | 7mer | 6mer | Context++ Score | miRNA |
|---|---|---|---|---|---|---|---|---|---|
| TGFBR1 | Transforming growth factor, beta receptor 1 | TGF-β | 4 | 1 | 3 | 0 | 2 | −0.26 | hsa-miR-548k |
| STAT1 | Signal transducer and activator of transcription 1 | JAK/STAT | 1 | 0 | 1 | 0 | 3 | −0.14 | hsa-miR-548k |
| NFKB1 | Nuclear factor of kappa light polypep-tide gene enhancer in B-cells | NF-κB | 1 | 0 | 0 | 1 | 1 | −0.11 | hsa-miR-548k |
| MAPK1 | Mitogen-activated protein kinase 1 | MAPK | 2 | 0 | 2 | 0 | 3 | −0.13 | hsa-miR-548k |
| STAT5A | Signal transducer and activator of transcription 5A | JAK/STAT | 2 | 0 | 1 | 1 | 0 | −0.03 | hsa-miR-548k |
| STAT5B | Signal transducer and activator of transcription 5B | JAK/STAT | 1 | 1 | 0 | 0 | 0 | −0.28 | hsa-miR-496 |
| TGFBR2 | Transforming growth factor beta receptor II | TGF-β | 1 | 1 | 0 | 0 | 3 | −0.14 | hsa-miR-496 |

Context++ scores (reflecting targeting strength).

miRNA binding site types (8mer, 7mer, etc.).

females and analyzed using miRNet to explore disease relevance. Among the upregulated miRNAs identified, hsa-miR-548k has been implicated in several pathological conditions, including diabetes mellitus [28], ischemic stroke [29], and a range of malignancies [54–58], based on comparative analyses between diseased populations and healthy controls. These associations suggest a potential role for hsa-miR-548k in vascular and metabolic dysregulation. Additionally, hsa-miR-190a-5p and hsa-miR-885-3p have been linked to the pathogenesis of various cancers, including glioblastoma, lung cancer, and mesothelioma [59–62], indicating their possible involvement in oncogenic pathways and tumor progression.

Among the downregulated miRNAs, hsa-miR-496 has been widely associated with various diseases, including type 2 diabetes [43], vascular disorders [48], stroke [49], and autoimmune conditions [[63–73]], primarily in studies comparing affected individuals with healthy controls. These associations are likely mediated through the regulation of key signaling pathways such as PI3K/Akt and TGF-β. Additionally, other downregulated miRNAs, including hsa-miR-641, miR-100-5p, and miR-133a-3p, have been linked to neurodegenerative diseases, cancer, and metabolic disorders [74–87], highlighting their roles in inflammation, cellular homeostasis, and disease progression. These findings emphasize the significance of miRNA dysregulation in the pathogenesis of chronic diseases and support their potential utility as diagnostic biomarkers and therapeutic targets. Further functional studies are necessary to validate these associations and elucidate the underlying molecular mechanisms.

Previous health assessments have demonstrated a disproportionately high burden of chronic diseases within the Marshallese population. Notably, an estimated 46.5% of Marshallese residents in Arkansas are affected by type 2 diabetes [88]. This prevalence is markedly elevated—approximately four times higher than that observed in the general U.S. population, highlighting a significant public health disparity [89].Our study revealed the presence of certain miRNAs with upregulated expression, including hsa-miR-548k in males and hsa-miR-496 in females that were associated with diabetes mellitus in diabetic populations as compared to general controls. Diabetes mellitus remains a major health concern in Marshallese communities and might accelerate comorbidities like obesity, cardiovascular diseases, and hypertension, all of which contribute to cognitive decline [12,13].

To further elucidate the underlying molecular mechanisms, we validated the predicted target genes of the differentially expressed miRNAs, hsa-miR-548k and hsa-miR-496, using the TargetScan database. These miRNAs are linked to genes involved in key signaling pathways implicated in the pathogenesis of diabetes and its complications, including insulin signaling, MAPK, Akt, JAK/STAT, TGF-β, and NF-κB—pathways that play central roles in regulating glucose metabolism, insulin

sensitivity, and inflammatory responses. Our findings are consistent with previous studies reporting associations between these miRNAs and diabetes mellitus [28,43]. Furthermore, hsa-miR-548k and hsa-miR-496 have also been associated with a range of related conditions, including obesity [43], vascular disorders [48], ischemic stroke [29,48,49,63], traumatic brain injury [69], and amyotrophic lateral sclerosis [69], based on analyses comparing diseased populations to healthy controls. These observations reinforce the relevance of these miRNAs in metabolic and neurological disease pathophysiology.

## Conclusion

Our findings suggest that hsa-miR-548k and hsa-miR-496 may serve as potential biomarkers for the early detection and progression of diabetes mellitus and its associated pathological complications. However, additional studies are required to elucidate the functional roles of these miRNAs in diabetic patients and to explore their associations with other disease states. Longitudinal monitoring of study participants will be essential to determine whether individuals with elevated levels of hsa-miR-548k and hsa-miR-496 are at increased risk for developing diabetes-related complications and comorbid conditions, as indicated by our bioinformatic analyses. Notably, these upregulated miRNAs have been previously linked to various diseases in other populations relative to healthy controls, suggesting that the observed miRNA alterations are not exclusive to the Marshallese population but are consistent with broader disease-associated expression patterns. Future research incorporating non-Marshallese control groups will be critical to distinguish between population-specific and disease-related miRNA signatures. While larger cohort studies are warranted to validate these findings, our results highlight the potential clinical utility of these miRNAs as diagnostic and therapeutic targets, particularly for addressing diabetes and its complications in Marshallese individuals.

## Limitations

While this study provides important preliminary insights, several limitations should be noted. First, the analysis was limited to Marshallese participants without inclusion of a non-Marshallese control group, restricting the generalizability of the findings. Second, the relatively small sample size may limit statistical power and the ability to detect subtle associations with clinical outcomes. Lastly, potential confounders such as diet, medication use, environmental exposures, and lifestyle factors were not controlled and may have influenced miRNA expression. Future studies should include larger, more diverse cohorts with appropriate controls and comprehensive metadata to validate and extend these findings.

## Materials and methods

### Study participants

The present study was conducted at the University of Arkansas for Medical Sciences (UAMS), its Northwest Regional Campus, and Institute for Community Health Innovation during the IRB approval period from February 15, 2023, to February 14, 2024 (IRB #273602). A total of 50 Marshallese participants (36 Females and 14 males) were enrolled based on eligibility of inclusion criteria and willingness to provide written informed consent during the study period. The sample size of 50 was based on the feasibility of recruitment within the Marshallese community in Arkansas, a small and underrepresented population with significant barriers to research participation, including limited health literacy and reluctance to consent. As this was a pilot study, no formal sample size calculation was performed. The research team consisted of doctors, nurses, and faculty/staff members. Participants were eligible if they were of Marshallese ethnicity, residing in Northwest Arkansas, aged 30–71 years, and of any gender. Individuals were excluded if they were not of Marshallese descent, lived outside of Northwest Arkansas, or were younger than 30 years old. Study materials were provided in both Marshallese and English. Recruitment was conducted by bilingual UAMS staff proficient in both Marshallese and English, spanning from February 15, 2023, to February 14, 2024. Data collection was managed using Research Electronic Data Capture (REDCap). Written informed consent was obtained from all participants.

## Sampling, RNA extraction, and sequencing

Among the total of 50 subjects (36 Females and 14 males) participated in the study, blood samples were obtained from 47 Marshallese participants (35 Females and 12 males) following written informed consent (IRB# 273602) with full clinical information available (Table 1). RNA was isolated with the PAXgene Blood miRNA Kit (Qiagen), adhering to the manufacturer's instructions. A total of 100 ng of total RNA was used to create miRNA next-generation sequencing libraries. The library preparation was done using the QIAseq miRNA Library Kit (QIAGEN). All samples were processed using the same library preparation protocol and sequencing platform. Samples were randomized across library preparation and sequencing batches to minimize technical variation. During the reverse transcription step, adapters and UMIs were introduced. The cDNA was then amplified via PCR (19 cycles), during which indices were also added. Post-PCR, the samples were purified. Library quality was assessed via capillary electrophoresis (Tape D1000). Based on the quality of the inserts and concentration measurements, the libraries were pooled in equimolar ratios. The pooled libraries were quantified with qPCR and subsequently sequenced on a NextSeq 2000 (Illumina Inc) according to the manufacturer's instructions (1x75, 2x10). Raw data were demultiplexed, and FASTQ files for each sample were generated with bcl2fastq2 software v2.20.0.422 (Illumina Inc). During data analysis, potential batch effects were assessed using principal component analysis (PCA) and found no evidence of strong batch-associated clustering. Therefore, no additional batch correction was applied.

## Read mapping and quantification of expression levels

All primary analyses were performed with CLC Genomics Server 23.0.5. The QIAseq miRNA Quantification workflow within the CLC Genomics Server, with standard parameters, was used to map reads to miRBase version 22. The process involved the following: (1) trimming of common sequences, UMIs, and adapters, and (2) filtering out reads shorter than 15 nt or longer than 55 nt. Reads were then deduplicated based on their UMI. Specifically, reads were grouped into UMI groups if they started at the same position relative to the end of the read where the UMI is ligated (i.e., Read2 for paired data), were from the same strand, and had identical UMIs. Singleton reads (groups that contain only 1 read) were merged into non-singleton groups if the UMI of the singleton could be converted to a UMI of a non-singleton group by introducing an SNP (the largest group was selected). Reads that failed to map to miRBase, either with perfect matches or as isomiRs (with a maximum of 2 mismatches and/or alternative start/end positions of up to 2 nt), were mapped to the human genome (GRCh38) with ENSEMBL GRCh38.98 annotation. This mapping was performed with the RNA-Seq Analysis workflow of the CLC Genomics Server with standard parameters.

## Unsupervised analysis

For unsupervised analyses, only miRNAs with a minimum of 10 counts summed across all samples were included. Normalized miRNA expression levels were used for principal component analysis (PCA). To perform hierarchical clustering, miRNAs with the highest variance across samples were selected via average linkage and Euclidean distance. A variance stabilizing transformation was applied to the raw count matrix via the *vst()* function from the R package DESeq2 (version 1.36.0), with variance calculated agnostically to the predefined groups (blind=TRUE). The software packages used for the data analysis were R [90], DeSeq2 [91], ggplot2 [92], rmdformats [93], and plotly [94]. The statistical analysis was performed, and the adjusted *P*-value cutoff for significance was set at *P*adj < 0.01.

## Differential expression analysis

Differential expression analysis was conducted with the Empirical Analysis of DGE algorithm in CLC Genomics Workbench 23.0.5, with default settings. This algorithm implements the Exact Test for 2-group comparisons, as developed by Robinson and Smyth (2008) [95] and incorporated into the EdgeR Bioconductor package [96]. Additionally, fold changes in protein expression in males versus females were also calculated and transformed on a log2 scale for normalization with a cutoff point of 1.5 for the

absolute log2 fold change in expression. Welch 2-sample t-test was performed between male and female groups, and *P*-values were calculated for differentially expressed proteins. Statistical analyses were performed with R 4.3.1.

### Dissecting miRNA-gene-target disease associations

The interactions of differentially expressed miRNAs were analyzed with miRNet 2.0 to dissect miRNA target interactions and functional associations through network-based visual analysis in the Human miRNA Disease Database [97]. To further investigate the potential functional implications of the differentially expressed miRNAs, hsa-miR-548k and hsa-miR-496, their predicted target genes were identified and validated using the TargetScan database (version 8.0) [98,99], which predicts biological targets of miRNAs by searching for the presence of conserved 8mer and 7mer sites that match the seed region of each miRNA.

## Supporting information

**S1 Fig. Sequencing analysis of samples.** (**A**) Number of reads per sample. The dotted vertical line indicates the mean number of reads. (**B**) For each sample, the median Phred quality score over all reads is shown at each read position. The colored bar classifies the Phred values into 3 quality categories: poor (red, 0 to < 20), medium (yellow, 20 to < 28) and good (green, 28 and higher). (**C**) Distribution of trimmed read length showed a distinct peak at approximately 22 nt, aligning with the expected miRNA length. (**D**) Read deduplication for each sample with the number of read pairs is plotted (dark blue) in relation to the number of collapsed UMI reads (light blue).
(TIF)

**S2 Fig. Mapping of reads to reference genome.** (**A**) Reads that failed to map to miRBase and other databases of small RNA were mapped to the ENSEMBL annotation. (**B**) Number of reads mapped to the reference genome split by gene element. (**C**) Proportion of reads per sample mapped to gene biotypes.
(TIF)

**S3 Fig. Hierarchical clustering top 100 miRNAs ranked by median absolute deviation.** A variance-stabilized transformation was performed on the raw count matrix. Each row represents 1 gene, and each column represents 1 sample. The color represents the difference of the count value to the row mean.
(TIF)

**S1 Table. Most significantly differentially expressed miRNAs in male versus female subjects.**
(DOCX)

**S1 Data. TargetScan8.0__miR-548k.Human.predicted_targets.**
(XLSX)

**S2 Data. TargetScan8.0__miR-496.Human.predicted_targets.**
(XLSX)

## Acknowledgments

We would like to thank Qiagen genomic services for the use of their equipment and technical expertise. We are grateful to Marshallese staff member Miss Lynda Ricklon for her help with recruitment, consent and language translation. Editorial support was provided by the Science Communication Group at the University of Arkansas for Medical Sciences.

## Author contributions

**Conceptualization:** Gohar Azhar, Jeanne Y. Wei.

**Data curation:** Gohar Azhar, Ambika Verma.

**Formal analysis:** Wei Zhang.

**Funding acquisition:** Pearl A. McElfish, Jeanne Y. Wei.

**Investigation:** Gohar Azhar, Jeanne Y. Wei.

**Methodology:** Gohar Azhar, Ambika Verma, Wei Zhang.

**Resources:** Gohar Azhar, Jeanne Y. Wei.

**Supervision:** Gohar Azhar, Jeanne Y. Wei.

**Visualization:** Gohar Azhar, Ambika Verma, Pankaj Patyal, Wei Zhang.

**Writing – original draft:** Gohar Azhar, Ambika Verma.

**Writing – review & editing:** Ambika Verma, Pankaj Patyal, Shakshi Sharma, Patricia E. Savary, Sheldon Riklon, Philmar Mendoza Kabua, Pearl A. McElfish, Jeanne Y. Wei.

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
