## [Decision Letter · Decision Letter 0]

Dear Dr. Azhar,

Thank you for submitting your manuscript to PLOS ONE. While your study on miRNA profiles in the Marshallese population is interesting, the manuscript requires revision before I can recommend it for publication.

The reviewers raised several key concerns that must be addressed. First, the study's objectives need clarification—you should clearly define whether the focus is on sex differences or disease associations. Second, the *Methods* section requires expansion to include detailed sample demographics, covariates like age and BMI, and justification for statistical thresholds. Third, the biological interpretation needs strengthening through pathway analysis and discussion of validated miRNA-disease relationships. Finally, ensure all figures are fully labeled and described.

Please also consider adding a limitations section addressing sample size and potential confounders. I recommend careful attention to these points, particularly the methodological and analytical issues, as they are essential to meet the PLOS ONE’s criteria for publication.

We look forward to receiving your revised manuscript.

Kind regards,

Sharif Moradi, PhD

Academic Editor

PLOS ONE

“This study was supported by grant R01MD013852 from the National Institutes of Health, and in part, by the Lyon Aging Research Program, Reynolds Institute on Aging, University of Arkansas for Medical Sciences, Little Rock, AR.”

4. Please update your submission to use the PLOS LaTeX template. The template and more information on our requirements for LaTeX submissions can be found at http://journals.plos.org/plosone/s/latex.

5. We note that your Data Availability Statement is currently as follows: [All relevant data are within the manuscript and its Supporting Information files.]

6. We notice that your supplementary tables are included in the manuscript file. Please remove them and upload them with the file type 'Supporting Information'. Please ensure that each Supporting Information file has a legend listed in the manuscript after the references list.

Reviewers' comments:

Reviewer's Responses to Questions

**Comments to the Author**

1. Is the manuscript technically sound, and do the data support the conclusions?

Reviewer #1: Yes

Reviewer #2: Partly

2. Has the statistical analysis been performed appropriately and rigorously?

Reviewer #1: Yes

Reviewer #2: Yes

3. Have the authors made all data underlying the findings in their manuscript fully available?

Reviewer #1: No

Reviewer #2: Yes

4. Is the manuscript presented in an intelligible fashion and written in standard English?

Reviewer #1: Yes

Reviewer #2: Yes

**Reviewer #1: ** This study presents novel miRNA sequencing data from an understudied population (Marshallese in Arkansas) with high rates of chronic diseases. While the technical execution is rigorous, several conceptual and analytical limitations reduce the impact of the findings. Some revisions are needed to clarify the study’s objectives, justify methodological choices, and strengthen biological interpretation.

Unclear Hypotheses

• The paper oscillates between two goals without clear prioritization:

o (a) Identifying sex-specific miRNA differences, and

(b) Linking miRNAs to chronic diseases (e.g., diabetes).

No statistical correlation is shown between miRNA expression and disease prevalence in participants (e.g., 46.5% diabetes rate).

• Suggestion: Refocus the manuscript on sex differences OR explicitly frame this as an exploratory biomarker discovery study.

• Sample Demographics: Missing details like age, sex distribution, and health status of participants, which could influence miRNA expression.

• Control Group: No mention of a non-Marshallese control group for comparative analysis. Are the miRNA differences unique to Marshallese people, or generalizable?

•

Arbitrary Statistical Thresholds

• The 1.5 log2FC cutoff for differential expression lacks justification. Most studies use |log2FC| > 1 (or 2) with FDR correction. The chosen threshold may inflate false negatives.

• No adjustment for covariates (age, BMI, medication) in DE analysis, despite known confounders in miRNA studies.

Lack of Mechanistic Insights

• The discussion lists disease associations (e.g., miR-548k and diabetes) but fails to:

o Perform pathway enrichment (e.g., KEGG/GO) to link miRNAs to biological processes.

o Validate targets experimentally or via published databases (e.g., TargetScan, miRDB).

Minor Revision:

Methods: Specify how batch effects were handled during sequencing/library prep.

Grammar: Some sentences are overly long (e.g., the final sentence could be split for readability).

Limitations: No discussion of limitations (e.g., small sample size, potential confounding variables like diet/environment).

• Legends (e.g., Fig 1, Fig 3) are overly brief. Include axes labels, sample sizes, and statistical thresholds in legends.

• Heatmaps (Fig 2B, 4B) lack color-scale interpretation (e.g., "red = high expression").

• "Variance-stabilized transformation" (line 143): Clarify why this was chosen over alternatives (e.g., TPM normalization).

• Table 1: Include units for "Mean" expression and clarify if values are normalized.

• Line 109: "Phred score >30" is standard; emphasize if any samples failed QC.

**Reviewer #2: ** In the manuscript titled “Differential microRNA profiling of the Marshallese population in Arkansas reveals a higher association with chronic diseases,” Azhar et al. performed a study to find how the miRNA expression profile differs in the Marshallese population with chronic diseases. Although the subject is interesting, several issues have to be addressed.

1-Although the authors noted the higher risk of the Marshallese community to chronic disease compared with the general population, no statistics were provided and no references were cited for this claim in the introduction.

2-Patient demographic and clinical characteristics need to be added to the manuscript.

3-Patients with chronic disease usually use multiple drugs. How did the authors account for the confounding effects of medications on miRNA expression?

4-The patient groups that were compared with each other and their sample sizes need to be clearly defined in the Methods section.

5-Why did the authors select 50 patients? How was the required sample size calculated for their study? This information should be added to the Methods section.

6-The term "chronic disease" refers to a wide range of conditions. The specific diseases under investigation should be introduced in the introduction. Also, it is common that patients with diabetes also suffer from other chronic diseases such as hypertension, cardiovascular disease, kidney disease, etc.

7-The duration of the study should be included in the Methods section.

8-Is gender considered a risk factor for chronic disease in the Marshallese community? It seems that the authors focused on comparing males and females throughout much of the study, but no background or rationale was provided in the introduction.

9-The different parts of Figure S2 (A, B, and C) should be described in the manuscript.

10-The results related to Figure 3B need to be described in the Results section, and the figure should be referenced in the text.

11-Much of the Discussion section simply repeats the results. The Discussion needs to be written more comprehensively and should include additional validated data regarding the role of the selected miRNAs in chronic disease and their involvement in disease pathogenesis.

**Do you want your identity to be public for this peer review?** For information about this choice, including consent withdrawal, please see our Privacy Policy

Reviewer #1: **Yes: ** Dr. Sara Taleahmad, Royan Institute, Tehran, Iran

Reviewer #2: No

---

## [Author Response · Author response to Decision Letter 1]

26 Jun 2025

Manuscript ID: PONE-D-25-14575

Title: Differential microRNA profiling of the Marshallese population in Arkansas reveals a higher association with chronic diseases

Journal: PLOS ONE

Dear Editor:

We sincerely thank you and the reviewers for your valuable feedback and the opportunity to revise and improve our manuscript. We appreciate the thoughtful and constructive comments that have helped us enhance the clarity, scientific rigor, and overall quality of the study.

We have carefully revised the manuscript in response to all the comments raised by the academic editor and reviewers. Below, we provide a point-by-point response to each concern, along with a description of how the manuscript was modified. All changes are highlighted in the revised manuscript—text highlighted in yellow corresponds to revisions made in response to Reviewer 1, while green highlights indicate changes relevant to Reviewer 2.

We hope that the revised manuscript now meets the journal’s expectations and standards for publication. Thank you again for your consideration.

Sincerely,

Gohar Azhar, M.D.

RESPONSE TO EDITOR

1. First, the study's objectives need clarification—you should clearly define whether the focus is on sex differences or disease associations.

Response: In response, we have revised the Introduction to clarify the study objectives with the justification of the inclusion of sex-related miRNA expression patterns and their potential associations with chronic diseases in the Marshallese population.

Please see revised Introduction, lines 77-87 and references (18, 19).

2. Second, the Methods section requires expansion to include detailed sample demographics, covariates like age and BMI, and justification for statistical thresholds.

Response: We have included a Table-1 with sample demographics and updated the Results in the revised manuscript.

Please see revised Results, lines 116-122 and Table-1.

We acknowledge that variables such as age and BMI, can be significant confounders in miRNA expression studies. However, due to the low health literacy and research hesitancy within the Marshallese population in Arkansas, we encountered incomplete medical records, which limited our ability to adjust for these covariates. We recognize this as a limitation and have addressed it in the revised Discussion section.

3. Third, the biological interpretation needs strengthening through pathway analysis and discussion of validated miRNA-disease relationships.

Response: As suggested, we have validated the target genes using TargetScan database and have incorporated the main findings in Table-3 and detailed information in Supporting file-1 and 2. We also revised the results and discussion sections accordingly.

Please see revised results, lines 271-287 and Table-3; revised Discussion, lines 358-363.

4. Finally, ensure all figures are fully labeled and described.

Response: We have made sure all the figures are correctly and fully labeled.

5. Please also consider adding a limitations section addressing sample size and potential confounders.

Response: We have added a limitations section which addresses the sample size and potential confounders as suggested.

6. I recommend careful attention to these points, particularly the methodological and analytical issues, as they are essential to meet the PLOS ONE’s criteria for publication.

Response: Thank you for your thoughtful recommendation. We fully acknowledge the importance of methodological and analytical rigor in meeting PLOS ONE’s publication criteria. In response, we have carefully revised our manuscript with particular attention to these areas as advised.

We appreciate your guidance and believe the revised manuscript now aligns with the journal’s standards.

RESPONSE TO REVIEWER 1

Reviewer #1: This study presents novel miRNA sequencing data from an understudied population (Marshallese in Arkansas) with high rates of chronic diseases. While the technical execution is rigorous, several conceptual and analytical limitations reduce the impact of the findings. Some revisions are needed to clarify the study’s objectives, justify methodological choices, and strengthen biological interpretation.

Response: We are deeply appreciative and grateful for all the thoughtful comments and helpful suggestions provided by Reviewer #1. We have revised the manuscript as suggested and believe that these suggested changes have strengthened the manuscript. The changes in the revised manuscript are highlighted in yellow.

Major Points:

1. Unclear Hypotheses: The paper oscillates between two goals without clear prioritization (a) Identifying sex-specific miRNA differences, and (b) Linking miRNAs to chronic diseases (e.g., diabetes). No statistical correlation is shown between miRNA expression and disease prevalence in participants (e.g., 46.5% diabetes rate). Suggestion: Refocus the manuscript on sex differences OR explicitly frame this as an exploratory biomarker discovery study.

Response: Thank you for this insightful feedback. In response, we have revised the Introduction to justify the inclusion of sex-related miRNA expression patterns and their potential associations with chronic diseases in the Marshallese population.

Please see revised Introduction, lines 77-87 and references (18, 19).

We acknowledge that no direct statistical correlation was performed between individual miRNA expression levels and disease prevalence. This limitation was primarily due to incomplete medical records, stemming from poor health literacy and research hesitancy within the Marshallese population in Arkansas, which restricted the availability of comprehensive clinical data.

2. Sample Demographics: Missing details like age, sex distribution, and health status of participants, which could influence miRNA expression.

Response: We have now included a Table-1 with sample demographics and updated the Results in the revised manuscript.

Please see revised Results, lines 116-122 and Table-1.

3. Control Group: No mention of a non-Marshallese control group for comparative analysis. Are the miRNA differences unique to Marshallese people, or generalizable?

Response: Thank you for this insightful comment. The present study focused specifically on profiling miRNA expression within the Marshallese population and did not incorporate a non-Marshallese control group for direct comparative analysis. However, several of the significantly upregulated miRNAs identified, have been previously reported in diverse disease contexts when comparing affected individuals to healthy controls. These consistent associations suggest that the miRNA alterations observed in our study are not exclusive to the Marshallese population but are indicative of broader disease-related expression patterns.

We have updated the Discussion to incorporate this point, lines 365-368.

4. Arbitrary Statistical Thresholds: The 1.5 log2FC cutoff for differential expression lacks justification. Most studies use |log2FC| > 1 (or 2) with FDR correction. The chosen threshold may inflate false negatives.

Response: We would like to clarify that we have also applied a threshold of |log2FC| > 1 along with FDR correction, as indicated in Figure 3 and Supplementary Table S1, which is consistent with commonly accepted standards in the field.

To further enhance the specificity of our analysis, we subsequently applied a more stringent cutoff of |log2FC| > 1.5 as indicated in Figure 4 and Table-2.

5. No adjustment for covariates (age, BMI, medication) in DE analysis, despite known confounders in miRNA studies.

Response: Thank you for raising this important point. We acknowledge that variables such as age, BMI, and medication use can be significant confounders in miRNA expression studies. However, due to the low health literacy and research hesitancy within the Marshallese population in Arkansas, we encountered incomplete medical records, which limited our ability to adjust for these covariates in the DE analysis. We recognize this as a limitation and have addressed it in the revised Discussion section.

Please see revised Discussion, lines 390-393.

6. Lack of Mechanistic Insights: The discussion lists disease associations (e.g., miR-548k and diabetes) but fails to Perform pathway enrichment (e.g., KEGG/GO) to link miRNAs to biological processes. Validate targets experimentally or via published databases (e.g., TargetScan, miRDB).

Response: As suggested, we have validated the target genes for miR-548k and mi-496 using TargetScan database and have incorporated the main findings in Table-3 and also revised the Results and discussion.

Please see revised results, lines 271-287 and Table-3; revised Discussion, lines 358-363.

Minor Points:

1. Methods: Specify how batch effects were handled during sequencing/library prep.

Response: As suggested, we have specified the batch effects handling in the updated Methods.

Please see revised methods, lines 419-422; 429-432.

2. Grammar: Some sentences are overly long (e.g., the final sentence could be split for readability).

Response: We have revised the manuscript by shortening long sentences where appropriate to improve clarity and readability.

3. Limitations: No discussion of limitations (e.g., small sample size, potential confounding variables like diet/environment).

Response: We have now included a limitations section at the end of Discussion.

Please see revised Discussion, lines 386-393.

4. Legends (e.g., Fig 1, Fig 3) are overly brief. Include axes labels, sample sizes, and statistical thresholds in legends.

Response: We have revised the legends for Figure 1 and Figure 3 as suggested.

5. Heatmaps (Fig 2B, 4B) lack color-scale interpretation (e.g., "red = high expression").

Response: We have added the color interpretation in the Figures legends 2B and 4B as suggested.

6. "Variance-stabilized transformation" (line 143): Clarify why this was chosen over alternatives (e.g., TPM normalization).

Response: We have added a clarification for this point in the results section of revised manuscript as suggested.

Please see revised results, lines 173-179.

7. Table 1: Include units for "Mean" expression and clarify if values are normalized.

Response: As suggested, we have included units for Mean expression in Table 1 (Now its Table-2) and also added clarification in the results of revised manuscript.

Please see revised results, lines 226-227 and Table-2.

8. Line 109: "Phred score >30" is standard; emphasize if any samples failed QC.

Response: Thank you for this question. In response, we have added an explanation in the results section of revised manuscript.

Please see revised results, lines 129-131.

RESPONSE TO REVIEWER 2

Reviewer #2: In the manuscript titled “Differential microRNA profiling of the Marshallese population in Arkansas reveals a higher association with chronic diseases,” Azhar et al. performed a study to find how the miRNA expression profile differs in the Marshallese population with chronic diseases. Although the subject is interesting, several issues have to be addressed.

Response: We are grateful to the reviewer #2 for the careful and insightful review of our manuscript. We have revised the manuscript and believe that these suggested changes have strengthened the manuscript. We have addressed Reviewer 2's comments, and the corresponding changes are highlighted in green in the revised manuscript. Changes highlighted in yellow indicate revisions that were common to both reviewers.

Major Points:

1. Although the authors noted the higher risk of the Marshallese community to chronic disease compared with the general population, no statistics were provided, and no references were cited for this claim in the introduction.

Response: Thank you for your helpful feedback. In response, we have revised the Introduction section and included the references as suggested.

Please see revised Introduction, lines 62-76 and references (1-17).

2. Patient demographic and clinical characteristics need to be added to the manuscript.

Response: We have now included a Table-1 with detailed information on Patient demographic and clinical characteristics in the revised manuscript.

Please see revised Results, lines 116-122 and Table-1 (highlighted in yellow).

3. Patients with chronic disease usually use multiple drugs. How did the authors account for the confounding effects of medications on miRNA expression?

Response: Thank you for raising this important point. We would like to mention that due to the low health literacy and research hesitancy within the Marshallese population in Arkansas, we encountered incomplete medical records, which limited our ability to study the effects of medications on miRNA expression. We acknowledge this as a limitation and have added a statement to the Discussion section.

Please see revised Discussion, lines 390-393 (highlighted in yellow).

4. The patient groups that were compared with each other and their sample sizes need to be clearly defined in the Methods section.

Response: We have revised the Methods section as suggested.

Please see revised methods, lines 414-416.

5. Why did the authors select 50 patients? How was the required sample size calculated for their study? This information should be added to the Methods section.

Response: We acknowledge the importance of justifying the sample size and have now addressed this point in the Methods section. We would like to mention that the sample size of 50 was based on the feasibility of recruitment within the Marshallese community in Arkansas, a small and underrepresented population with significant barriers to research participation, including limited health literacy and reluctance to consent. As this was a pilot study, no formal sample size calculation was performed.

Please see revised methods, lines 399-404.

6. The term "chronic disease" refers to a wide range of conditions. The specific diseases under investigation should be introduced in the introduction. Also, it is common that patients with diabetes also suffer from other chronic diseases such as hypertension, cardiovascular disease, kidney disease, etc.

Response: As suggested, we have revised the Introduction to explicitly define the chronic diseases associated with our study.

Please see revised introduction, lines 62-76 and references 1-17.

7. The duration of the study should be included in the Methods section.

Response: We have added the duration of the study in Methods section of revised manuscript as suggested.

Please see revised methods, line 398.

8. Is gender considered a risk factor for chronic disease in the Marshallese community? It seems that the authors focused on comparing males and females throughout much of the study, but no background or rationale was provided in the introduction.

Response: Thank you for your thoughtful observation. In response, we have revised the Introduction to include background on the relevance of gender as a biological variable in chronic disease research.

Please see revised introduction, lines 77-87 and references 18, 19 (highlighted in yellow).

9. The different parts of Figure S2 (A, B, and C) should be described in the manuscript.

Response: Thank you for your helpful comment. We have now revised the Results to include a specific description of each panel in Figure S2 (A, B and C).

Please see revised results section- Mapping statistics, lines 151-162.

10. The results related to Figure 3B need to be described in the Results section, and the figure should be referenced in the text.

Response: We have now revised the results to include a detailed description of the findings shown in Figure 3B and have referenced the figure appropriately within the Results section.

Please see revised results, lines 203-207.

11. Much of the Discussion section simply repeats the results. The Discussion needs to be written more comprehensively and should include additional validated data regarding the role of the selected miRNAs in chronic disease and their involvement in disease pathogenesis.

Response: We appreciate the reviewer’s insightful feedback regarding the Discussion section. In response, we have substantially revised the Discussion to reduce repetition of the Results and provide a more comprehensive, mechanistic interpretation of our findings.

---

## [Editor Report · Decision Letter 1]

Differential microRNA profiling of the Marshallese population in Arkansas reveals a higher association with chronic diseases

PONE-D-25-14575R1

Dear Dr. Azhar,

I am pleased to inform you that your manuscript has been judged scientifically suitable for publication and will be formally accepted for publication once it meets all technical requirements.

Kind regards,

Sharif Moradi, Ph.D.

Academic Editor

PLOS ONE
---

## [Editor Report · Acceptance letter]

PONE-D-25-14575R1

PLOS ONE

Dear Dr. Azhar,

I'm pleased to inform you that your manuscript has been deemed suitable for publication in PLOS ONE. Congratulations! Your manuscript is now being handed over to our production team.

Kind regards,

on behalf of

Dr. Sharif Moradi

Academic Editor

PLOS ONE